# Heterogeneity of Endocrinologic and Metabolic Parameters in Reproductive Age Polycystic Ovary Syndrome (PCOS) Women Concerning the Severity of Hyperandrogenemia—A New Insight on Syndrome Pathogenesis

**DOI:** 10.3390/ijerph17249291

**Published:** 2020-12-11

**Authors:** Katarzyna Ozegowska, Marcin Korman, Agnieszka Szmyt, Leszek Pawelczyk

**Affiliations:** Department of Infertility and Reproductive Endocrinology, Poznan University of Medical Sciences, 60-535 Poznań, Poland; mkorman@poczta.onet.pl (M.K.); agusia.szmyt@gmail.com (A.S.); leszek.a.pawelczyk@gmail.com (L.P.)

**Keywords:** polycystic ovary syndrome, hyperandrogenism, free testosterone, metabolic disturbances

## Abstract

Background: Polycystic ovary syndrome (PCOS) is characterized by hyperandrogenism, anovulation, infertility, obesity, and insulin resistance, which results in increased concentrations of testosterone (T), which disturbs follicular growth and ovulation. This study aimed to assess PCOS women’s clinical, endocrinological, and metabolic parameters concerning hyperandrogenism severity. Results: 314 women (mean age 27.3 ± 4.6; mean body mass index (BMI) 25.7 ± 5.6) with PCOS, were divided into terciles according to T concentrations: <0.64 ng/mL (group 1), 0.64 to 0.84 ng/mL (Group 2) and >0.84 ng/mL (group 3). The mean concentration of T in all women was 0.59 ng/mL and correlated negatively with the number of menstrual cycles per year (MPY) (*r* = −0.36; *p* < 0.0001) and positively with Ferriman-Gallway score (FG) (*r* = 0.33; *p* < 0.0001), luteinizing hormone (LH) (*r* = 0.19; *p* < 0.0001) and dehydroepiandrosterone sulfate (DHEAS) (*r* = 0.52; *p* < 0.0001). Positive correlation between BMI and hirsutism (*r* = 0.16; *p* < 0.0001), total cholesterol (TC) (*r* = 0.18; *p* < 0.0001), low-density lipoprotein (LDL) (*r* = 0.29; *p* < 0.0001), and triglycerides (TG) (*r* = 0.40; *p* < 0.0001) was demonstrated. The division into subgroups confirmed the lowest MPY, highest LH, and hirsutism in group 3. BMI, insulin sensitivity indices, and lipid profile parameters were not different between the three T subgroups. Conclusions: We found no correlation between testosterone levels and insulin sensitivity or dyslipidemia in women with PCOS. Metabolic abnormalities may contribute more significantly than hyperandrogenemia to PCOS development.

## 1. Introduction

Polycystic ovary syndrome (PCOS) is one of the most common endocrinopathies affecting 4 to 21% of women of reproductive age, depending on the diagnostic criteria. Prevalence of the syndrome varies according to different geographic regions, but mainly because of various diagnostic consensus used. Ranges 5% to 10% according to NIH 1990 criteria; 10% to 15% according to the AE-PCOS 2006 criteria, and 6% to 21% by ESHRE/ASRM 2003 criteria are applied [1,2,3,4,5,6,7,8].

Hyperandrogenism, chronic anovulation, infertility, and obesity characterize PCOS. It is also associated with insulin resistance and hyperinsulinemia, leading to the development of type 2 diabetes and cardiovascular disease [9,10,11,12,13,14]. The pathophysiology of PCOS is still not fully understood. Patients with PCOS suffer not only from reproductive dysfunction [15,16,17,18], but also are at increased risk of long-term metabolic and cardiovascular health problems such as dyslipidemia [19,20], gestational diabetes [21], type 2 diabetes [22], hypertension [9,23,24], and atherosclerosis s [25,26]. The most prominent endocrine PCOS features include insulin resistance (IR) and compensatory hyperinsulinemia. Carmina et al., reported that the prevalence of IR range from 68% to 76% according to glucose tolerance test [27]. In a large study on untreated patients with PCOS, the prevalence of IR was approximately 64% according to the HOMA-IR measurement [28]. Insulin resistance is observed in both obese and non-obese women with PCOS. Legro et al., reported a higher prevalence of insulin resistance in obese (64%) than in non-obese (20%) patients [29]. Most evidence indicates that hyperinsulinemia causes hyperandrogenism (HA) rather than its converse. Insulin in higher concentration also inhibits sex hormone-binding globulin (SHBG) production in the liver, resulting in increased free testosterone [11,22,30].

Consequently, androgen excess has an unfavorable effect on follicular growth and ovulation and may compromise oocytes and embryos [31,32]. Hyperandrogenemia may lead to dysfunction of the corpus luteum, reduced progesterone production, and finally to impaired endometrial receptivity [33,34]. However, despite continuous research, there are still controversies regarding PCOS’s etiology, pathogenesis, and phenotypic spectrum.

This study aimed to assess PCOS women’s clinical, endocrinological, and metabolic parameters concerning hyperandrogenism severity.

## 2. Materials and Methods

### 2.1. Study Subjects

Three hundred fourteen women of reproductive age (mean age 27.3 ± 4.6; mean BMI 25.7 ± 5.6) with diagnosed PCOS according to the Rotterdam criteria [13] were recruited to the study. We defined PCOS as oligo-or amenorrhea, HA (hirsutism or acne), and/or hyperandrogenemia (increased total serum testosterone (T) >0.8 ng/mL). The ovarian transvaginal ultrasonographic evaluation demonstrated that all the women studied had multiple (more than 12) subcapsular antral follicles measuring up to 10 mm of diameter, i.e., all had classic sonographic characteristics of polycystic ovaries. All recruited subjects were euthyroid with no evidence of other endocrinopathies, including hyperprolactinemia, Cushing’s syndrome, congenital adrenal hyperplasia, or androgen-secreting tumors. No treatment, neither oral contraceptives, nor other hormones, and drugs that could affect carbohydrate metabolism were taken by the participants during the study and two months before the study’s commencement.

All women underwent clinical assessment in the early follicular phase. Physical examination, including the measurement of weight, height, body mass index (BMI), waist and hip circumferences, was performed, and hirsutism was assessed with Ferriman–Gallwey score (FG). Blood samples were taken from patients during the early follicular phase between 7 a.m. and 8 a.m., after 12-h fasting.

### 2.2. Assays

Serum levels of insulin were determined by standard ELISA (Enzyme Test Insulin; Boehringer Mannheim, Mannheim, Germany). T, LH, FSH, plasma glucose, and prolactin were measured with specific chemiluminescence assays (Chiron Diagnostics GmbH, Fernwald, Germany). Dehydroepiandrosterone sulfate (DHEAS) and SHBG were measured with radioimmunoassays: (Orion Diagnostica, Espoo, Finland). Total cholesterol (TC), HDL, LDL, and triglycerides (TG) were measured with an enzymatic colorimetric method on Roche Cobas Integra 400. Sampled sera were stored at −20 °C for subsequent analysis. Free androgen index (FTI) was derived using formula: T (nmol/L) × 100/SHBG (nmol/L). The fasting glucose to insulin ratio (FGIR) was calculated as proposed by Legro [35]. Quantitative insulin sensitivity check index (QUICKI = 1/[log (fasting insulin) + log (fasting glucose)] was calculated as described by Katz et al. [36]. HOMA-IR was calculated using the formula: HOMA-IR = [glucose (nmol/L) × insulin (µU/mL)/22.5], using fasting values [37].

The intra-and inter-assay coefficients of variation were below 10% for all assays performed.

### 2.3. Studied Subgroups

The studied population was divided into terciles according to T concentrations. The lowest tercile comprised patients with T level <0.64 ng/mL (group 1), the middle tercile ranged from 0.64 to 0.84 ng/mL (group 2), and the highest one was above 0.84 ng/mL (group 3). The testosterone level in our laboratory that is proposed to be within the normal range for women of reproductive age is 0.06–0.82 ng/mL.

All laboratory measurements were performed in the certified central laboratory of the Gynecologic Obstetrical University Hospital in Poznan.

The Institutional Ethical Committee of Poznan University of Medical Sciences approved this retrospective study protocol (349/14).

### 2.4. Statistical Analysis

Statistical analyses were performed using Statistica version 10. PL software (StatSoft, Inc., Tulsa, OK, USA) and MedCalc Statistical Software version 16.4. 3 (MedCalc Software bv., Ostend, Belgium; https://www.medcalc.org; 2016).

Testing for normality of data distribution was performed using the Kolmogorov–Smirnov and D’Agostino Pearson tests. The Kruskal-Wallis was used to determine the differences between the studied groups. Data were presented as median (IQR 25–75). The Spearman’s Rank Correlation Coefficient was used to discover the strength of a link between two sets of data. The *p*-value of less than 0.05 was considered significant.

## 3. Results

We examined 694 patients diagnosed with PCOS according to Rotterdam criteria [13], with median age 27.0 (24.0–31.0) and median T level 0.54 (0.39–0.75) (Table 1).

Table 2 presents the characteristic of the study group according to their testosterone concentration (terciles). The worth underlying finding is that the studied subgroups did not differ concerning their age and BMI. However, we observed the tendency to higher BMI in the group with the highest T. This characteristic initially did not indicate the relation between the patients’ androgenemia and metabolic status in these subgroups of PCOS patients.

We performed intergroup comparisons of FTI between terciles that revealed significant differences, with expected the highest level in group 3 (*p* < 0.05) (Figure 1a, Table 2).

Further analysis of the studied subpopulations divided into T terciles revealed worthy of note observations: we confirmed significant differences in menstrual cyclicity, with the lowest number of MPY in group 3 (Figure 1b; Table 2), LH (Figure 1c, Table 2) and hirsutism (Figure 1d, Table 2). On the other hand, BMI (Figure 1e, Table 2), insulin sensitivity indices (Figure 1f,g, Table 2), and lipid profile parameters (Table 2) were not different between the three T subgroups.

In the third table we present the correlation between studied parameters in the whole group. The mean T concentration in all PCOS subjects was 0.59 (0.27) and correlated negatively with the number of menstrual cycles per year (MPY) (Spearman correlation *r* = −0.36; *p* < 0.0001) and positively with FG score (*r* = 0.33; *p* < 0.0001), LH (*r* = 0.19; *p* < 0.0001) and with DHEAS (*r* = 0.52; *p* < 0.0001). Likewise, also FTI correlated with MPY (*r* = −0.24; *p* < 0.001), FG (*r* = 0.34; *p* < 0.0001) and DHEAS (*r* = 0.36; *p* < 0.0001) but additively showed positive correlation with BMI (*r* = 0.38; *p* < 0.0001) and TG (*r* = 0.26; *p* < 0.0001), LDL (*r* = 0.09; *p* = 0.02), TC/HDL ratio (*r* = 0.27; *p* < 0.0001) and negative with HDL (*r* = −0.27; *p* < 0.0001), QUICKI (*r* = −0.27; *p* < 0.001) and FGIR (*r* = −0.24; *p* < 0.0001). Moreover, a positive correlation between BMI and hirsutism (*r* = 0.16; *p* < 0.0001), TC (*r* = 0.18; *p* < 0.0001), LDL (*r* = 0.29; *p* < 0.0001), TC/HDL (*r* = 0.49; *p* < 0.0001) and TG (*r* = 0.40; *p* < 0.0001) was demonstrated. Negative correlations were shown between BMI and LH (*r* = −0.18; *p* < 0.0001), FGIR (*r* = −0.45; *p* < 0.0001), QUICKI (*r* = −0.556; *p* < 0.0001), as well as HDL (*r* = −0.45; *p* < 0.0001) levels (Table 3).

Subsequently we presented analysis of relationships between measured parameters within subgroups. These data are shown in Table 4, Table 5 and Table 6 (For the clarity we enclosed only significantly different data in the enclosed tables). In the Table 4 we present correlation in the selected parameters in Group 1. Obviously, there is a positive correlation between T and FTI (*r* = 0.586; *p* < 0.0001). BMI had strong, significant, positive correlation with TC/HDL (*r* = 0.586; *p* < 0.0001); HOMA-IR (*r* = 0.548; *p* < 0.0001), TG/HDL (*r =* −0.511; *p* < 0.0001) and negative correlation with FGIR (*r* = −0.518; *p* < 0.0001)) and QUICKI (*r* = −0.545; *p* < 0.0001). FGIR correlated significantly with QUICKI (*r* = 0.945; *p* < 0.0001). We also noticed significant negative correlation between QUICKI and TC/HDL (*r* = −0.513; *p* < 0.0001). TC correlated positively with LDL (*r* = 0.839; *p* < 0.0001). TC/HDL correlated obviously with HDL (*r* = −0.778; *p* < 0.0001), LDL (*r* = 0.686; *p* < 0.0001); TG (*r* = 0.636; *p* < 0.0001); HOMA-IR (*r* = 0.518, *p* < 0.0001), TG/HDL (*r* = 0.805; *p* < 0.0001). TG/HDL correlated positively with TG (*r* = 0.905; *p* < 0.0001) and negatively with HDL (*r* = −0.764; *p* < 0.0001).

In the Table 5 we present the results for the Group 2. Similarly, to the Group 1 we see that the strong correlations are observed mostly within metabolic parameters. BMI correlates significantly with FGIR (*r* = −0.58; *p* < 0.0001); QUICKI (*r* = −0.62; *p* < 0.0001), TC/HDL (*r* = 0.515; *p* < 0.0001), HOMA-IR (*r* = 0.952; *p* < 0.0001); TG/HDL (*r* = 0.535; *p* < 0.0001). FGIR correlated positively with QUCIKI (*r* = 0.953; *p* < 0.0001) and negatively with HOMA-IR (*r* = −0.952; *p* < 0.0001). HOMA-IR had strong correlation with QUICKI (*r* = −1.0, *p* < 0.0001) and TG/HDL (*r* = 0.519; *p* < 0.0001).

In the Table 6 we show the results of Group 3 with the highest testosterone levels. BMI strongly had significant positive correlation with TC/HDL (*r* = 0.544; *p* < 0.0001), HOMA-IR (*r* = 0.637; *p* < 0.0001); TG/HDL (*r* = 0.544; *p* < 0.0001) and negative with FGIR (*r* = −0.65; *p* < 0.0001), QUICKI (*r* = −0.643; *p* < 0.0001), HDL (*r* = −0.59; *p* < 0.0001). FGIR correlated positively with QUICKI (*r* = 0.908; *p* < 0.001) and negatively with HOMA-IR (*r* = −0.907; *p* < 0.0001) and TG/HDL (*r* = −0.5; *p* < 0.0001). There was also a correlation between HOMA-IR and HDL (*r* = −0.517; *p* < 0.0001), TC/HDL (*r* = 0.504; *p* < 0.0001) and TG/HDL (*p* = 0.529; *r* < 0.001).

## 4. Discussion

PCOS, also named “functional hyperandrogenism,” is regarded as androgen-induced ovarian dysfunction because these are androgens responsible for the clinical features of the syndrome [38]. On the ovarian level, they interfere with folliculogenesis and lead to the characteristic appearance of ovaries [39]. Numerous papers confirm that a higher incidence of IR with compensatory hyperinsulinemia is a major contributor to the overproduction of male sex steroids [11,40]. Insulin resistance is an essential regulatory factor of ovarian steroidogenesis and, in concert with LH, through cytochrome P450c17α, acts as a gonadotrophic hormone [41]. Moreover, hyperinsulinemia increases bio-available serum testosterone by inhibiting the hepatic synthesis of sex hormone-binding globulin (SHBG) [41,42].

In PCOS girls during puberty, physiological hyperinsulinemia may induce both ovarian hyperandrogenemia and anovulation. These adverse effects are exaggerated by higher than normal insulin levels, probably due to genetic predisposition or excessive weight gain (or both). Increased secretion of insulin and reduced insulin sensitivity were found in obese adolescent girls with PCOS clinical features compared to weight-matched control subjects [43,44]. These pathognomonic characteristics persist till adulthood. In a study assessing the large population of 1212 women with different PCOS phenotypes, obese subjects with mild disease (anovulation and polycystic ovaries without HA) were more insulin resistant than healthy BMI-matched controls and had similar IR as women with HA.

On the other hand, normal-weight women with that phenotype did not differ from controls in IR markers [45]. In our study, we did notice a significantly lower rate of MPY in the groups with middle and highest groups of T, but there were no differences between all the groups according to insulin sensitivity parameters (fasting insulin, HOMA-IR, QUICKI, and FGIR). Moreover, we could not find any strong correlations between MPR and other examined parameters.

In our analysis, we observed that in the third tercile of patients, DHEA-S concentrations were significantly the highest. DHEA-S in the whole population had a strong positive correlation with T, correlated negatively with cholesterol/HDL ratio. Moreover, in the whole studied population, levels of DHEAS were in negative correlation with BMI. An increase in adrenal androgen production is present in 20–60% of women with PCOS [46,47,48]. Increased 17-α-hydroxylase/17-20 lyase (P450 17α) is probably responsible for enhanced androgen biosynthesis in PCOS [49,50]. The polymorphism in the regulatory region of CYP17, a gene that codes for cytochrome P450c17-α, probably up-regulates its expression and may result in augmented androgens production [51]. Numerous studies assessing the association between CYP17 polymorphism and PCOS risk have brought conflicting results for the last two decades. Finally, a recent meta-analysis, which processed all published case-control studies, did not demonstrate such a correlation in the overall study. However, limiting the analysis to the studies within Hardy–Weinberg equilibrium (assuming that both allele and genotype frequencies in a population remain constant) significantly increased risk of CYP17 polymorphism in PCOS patients was proved [52,53].

Lipid abnormalities in PCOS have been initially attributed to the presence of insulin resistance [19,40]. However, some studies report a relation between hyperandrogenemia and dyslipidemia, although the mechanism is not precise [54]. High androgen levels probably exert that effect by working directly at the liver, or they alter body composition by favoring central adiposity [55,56]. In our study, although we found a correlation of FTI with BMI, insulin sensitivity indices on lipid profile parameters in the lowest T group, a similar relation was not fully demonstrated in the middle and highest T tercile. We noticed that it was not the testosterone level itself, but metabolic parameters such as BMI, HOMA-IR, QUICKI that correlated with lipid profile. In our study, levels of testosterone did not correlate with the intensity of metabolic disturbances.

Calculation of LH concentrations revealed statistically significant differences between studied subgroups, revealing that high T tercile patients had the highest LH levels. This phenomenon might be explained by reduced steroid hormone negative feedback on LH secretion because of androgen excess, enhancing hypothalamic gonadotropin releasing-hormone (GnRH) pulsatile release and causes the increase of LH pulse frequency [57,58]. This neuroendocrine abnormality occurs in adolescent girls and probably requires a genetic component or depends on androgen excess [59]. It was proven that exposure to elevated levels of androgens during puberty in female rhesus monkeys changes the neural drive to the reproductive axis. It increases LH pulsatile secretion and LH response to GnRH [60,61]. The full expression of PCOS’s clinical characteristics depends on maturational changes during puberty when a regular alteration in LH pattern may unveil hypersecretion of androgens in adolescent girls with polycystic ovaries [62,63,64]. Some authors point sexually dimorphic pattern of LH release, which more closely resembles that of men or women with congenital virilizing disorders, e.g., girls with CAH have a predisposition to the development of clinical and biochemical features of PCOS [65]. In the study investigating an association of LH β-subunit gene variants with PCOS, the authors demonstrated that cases harboring variant alleles had higher mean LH levels.

Moreover, the prevalence of LH genes variants was lower in obese than lean PCOS women [66]. In our observations, LH correlated negatively with BMI, but those were not strong correlations. Furthermore, there was no strong correlation between LH and T, FTI, and DHEA-S, as well as with any metabolic parameters.

In summary, PCOS is the commonest cause of menstrual irregularity and hyperandrogenemia. It is often associated with many metabolic dysfunctions that were postulated to be responsible for the pathogenesis of the disease. Insulin resistance, central to this disorder, might be regarded as exaggerating standard metabolic alteration during puberty, which is further exacerbated by weight gain. From another perspective, there is evidence for primary abnormality in ovarian folliculogenesis and androgen production manifested in puberty, but their origins may reach early infancy or even the fetal period. Although PCOS etiology is still uncertain, preceding observations suggest that there may exist many different pathways leading to the development of that syndrome.

The advantage of the study is the group size and homogeneity of the studied population. Not using the hyperinsulinemic-euglycemic clamp is the study’s limitation when speculating about hyperinsulinemia effect on the variables studied in this paper. However, the study of Pisprasert et al., indicated that insulin sensitivity indices based on glucose and insulin levels work well in the studies where we examine a homogenous group of patients (same gender, same race population) [67].

## 5. Conclusions

Our study confirms that PCOS is a very heterogeneous syndrome manifesting with diverse clinical, endocrinological, and metabolic phenotypes. On a population of white Caucasian PCOS women, we demonstrated that hyperandrogenemia correlates with severity of derangements in clinical parameters, that all predispose to the development of the metabolic syndrome, which we have described already in our previous papers [68,69].

Interestingly, subgroups’ analysis revealed no correlation between androgen levels and insulin sensitivity/dyslipidemia in women with high/very high levels of testosterone. We speculate that metabolic abnormalities may contribute more significantly to PCOS development in women’s subpopulation with no aberrant steroidogenesis and less predominant androgenization.

## Figures and Tables

**Figure 1 ijerph-17-09291-f001:**
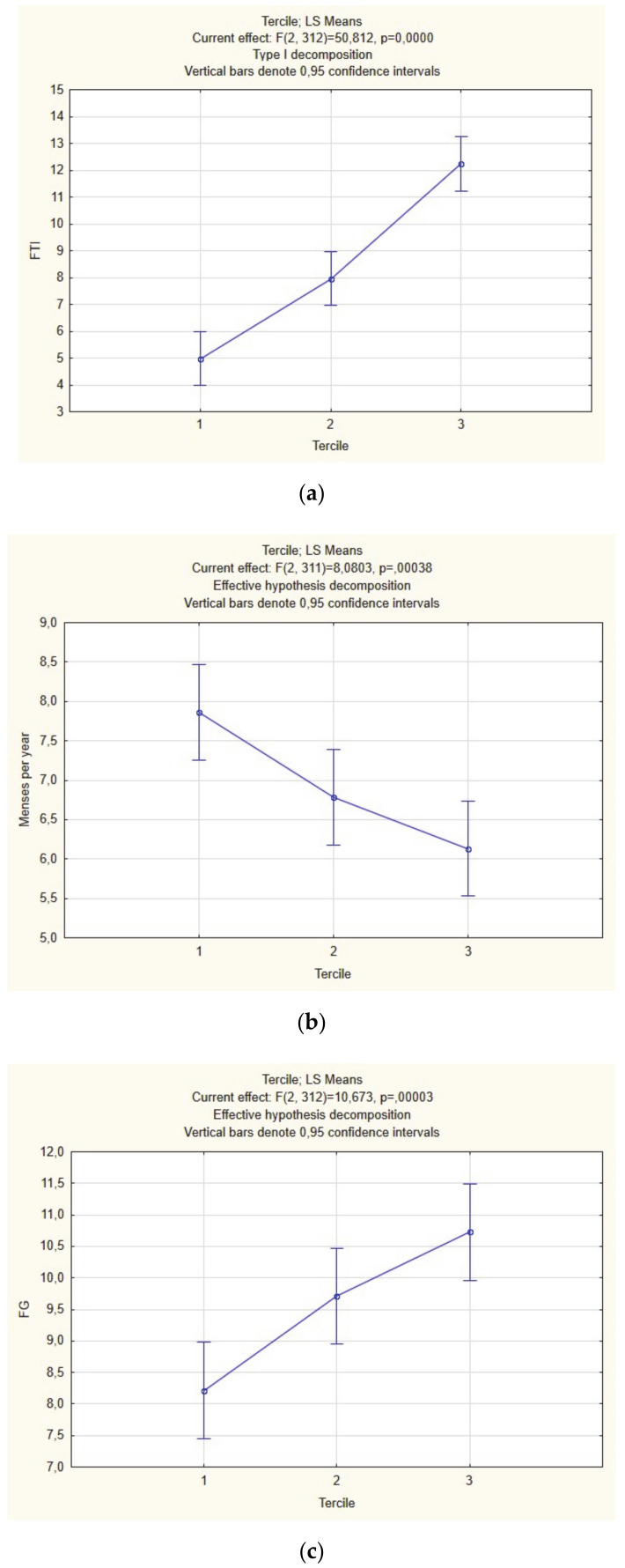
Intergroup comparison between terciles of free testosterone index (FTI) (**a**), menses per year (MPY) (**b**), luteinizing hormone (LH) (**c**), Ferrimann–Gallway score (FG) (**d**), body-mass index (BMI) (**e**), QUICKI (**f**), FGIR (**g**) (ANOVA).

**Table 1 ijerph-17-09291-t001:** Characteristics of the whole studied population.

Variable	PCOS (*n* = 694)
Age (years)	27.0 (24.0–31.0)
T (ng/mL)	0.54 (0.39–0.75)
Insulin	7.86 (5.30–11.97)
Glucose	88.0 (82.0–94.2)
HOMA-IR	1.74 (1.10–2.68)
FTI	4.90 (2.738.23)
MPR	7.0 (5.0–9.0)
FG	9.0 (4.0–11.0)
BMI (kg/m2)	24.2 (21.3–29.1)
LH (mIU/mL)	7.94 (5.49–11.42)
DHEA-S (umol/L)	6.01 (3.49–8.85)
FGIR	11.07 (7.42–16.27)
QUCIKI	0.35 (0.33–0.38)
CHOL (mg/dL)	182.0 (161.6–205.0)
HDL (mg/dL)	60.45 (49.90–72.00)
TG (mg/dL)	76.5 (58.0–110.0)
LDL (mg/dL)	103.0 (85.0–123.6)
TC/HDL	2.98 (2.43–3.66)
TG/HDL	1.21 (0.86–2.10)

Data are presented as median (IQR 25-75). Abbreviations: BMI—body mass index; CHOL—cholesterol; DHEA-S-dehydroepiandrostendione sulfate; FG—Ferriman–Gallway score; FGIR—fasting glucose to insulin ratio; FTI—free testosterone index; HDL—high-density lipoprotein cholesterol; HOMA-IR—Homeostatic Model Assessment for Insulin Resistance; LH-luteinizing hormone; LDL—low-density lipoprotein cholesterol; MPR—menstrual cycles per year; QUICKI—quantitative insulin sensitivity check index; TC-total cholesterol; TC/HDL—total cholesterol to high-density lipoprotein cholesterol ratio; TG—triglycerides; T—total testosterone.

**Table 2 ijerph-17-09291-t002:** Characteristics of the study groups.

Variable	Group 1 (*n* = 294)	Group 2 (*n* = 261)	Group 3 (*n* = 139)	*p* ^#^
Age (years)	27.5 (24.0–31.0)	27.0 (24.0–31.0)	27.0 (24,0–30.0)	0.66
Insulin	7.6 (5.3–11.0)	8.4 (5.4–12.6)	7.8 (5.2–13.2)	0.25
Glucose	88.0 (82.0–94.0)	88.6 (83.0–95.0)	89.0 (81.0–95.0)	0.46
HOMA-IR	1.7 (1.1–2.4)	1.8 (1.1–2.8)	1.7 (1.1–2.9)	0.24
FTI	2.7 (1.6–4.5)	5.7 (3.9–8.6)	10.1 (6.1–15.5)	<0.000001
MPR	9.6 (7.0–12.0)	7.1 (5.0–9.0)	7.0 (4.0–8.0)	<0.00001
FG	5.0 (1.0–10.0)	9.0 (7.0–11.0)	10.0 (8.0–12.0)	<0.000001
BMI (kg/m2)	23.7 (20.8–28.2)	25.0 (21.6–30.0)	24.1 (21.6–30.1)	0.03
LH (mIU/mL)	6.8 (5.2–9.9)	8.4 (5.7–11.8)	9.6 (6.4–13.3)	<0.000001
DHEA-S (umol/L)	4.4 (1.1–6.50	6.8 (4.0–9.2)	9.7 (6.8–12.4)	<0.000001
FGIR	11.6 (8.1–16.3)	10.7 (7.0–15.9)	10.9 (6.9–16.6)	0.23
QUCIKI	0.2 (0.3–0.4)	0.4 (0.3–0.4)	0.4 (0.3–0.4)	0.22
CHOL (mg/dL)	178.8 (157.0–204.0)	184.4 (165.3–204.3)	184.9 (164.1–208.0)	0.1
HDL (mg/dL)	60.6 (49.0–72.0)	60.2 (49.8–72.5)	60.6 (50.1–71.3)	0.9
TG (mg/dL)	73.0 (57.7–105.5)	78.3 (60.2–117.2)	77.7 (55.9–105.3)	0.49
LDL (mg/dL)	101.0 (85.8–122.0)	103.0 (85.0–122.0)	108.0 (79.3–126.4)	0.75
TC/HDL	2.95 (2.4–3.6)	3.0 (2.4–3.7)	3.0 (2.4–3.6)	0.56
TG/HDL	1.2 (0.9–2.0)	1.3 (0.9–2.2)	1.2 (0.9–2.0)	0.54

^#^ Kruskal–Wallis test; data are presented as median (IQR 25-75). Abbreviations: BMI—body mass index; CHOL—cholesterol; DHEA-S—dehydroepiandrostendione sulfate; FG—Ferriman–Gallway score; FGIR—fasting glucose to insulin ratio; FTI—free testosterone index; HDL—high-density lipoprotein cholesterol; HOMA-IR—Homeostatic Model Assessment for Insulin Resistance; LH-luteinizing hormone; LDL—low-density lipoprotein cholesterol; MPR—menstrual cycles per year; QUICKI-quantitative insulin sensitivity check index; TC-total cholesterol; TC/HDL—total cholesterol to high-density lipoprotein cholesterol ratio; TG—triglycerides; T—total testosterone.

**Table 3 ijerph-17-09291-t003:** Correlation between studied parameters in the whole polycystic ovary syndrome (PCOS) group.

Variable	#	T	FTI	MPR	FG	BMI	LH	DHEA	FGIR	QUICKI	Chol	HDL	LDL	TC/HDL	TG	HOMA-IR	TG/HDL
T (ng/mL)	*r*		0.626	−0.355	0.326	0.105	0.188	0.518	0.001	−0.021	0.054	−0.017	0.012	0.061	0.046	0.05	0.035
	*p*		<0.0001	<0.0001	<0.0001	0.0058	<0.0001	<0.0001	0.97	0.5914	0.1577	0.6614	0.7474	0.1111	0.2247	0.19	0.36
FTI	*r*	0.626		−0.238	0.304	0.377	0.012	0.366	−0.204	−0.271	0.05	−0.266	0.089	0.271	0.258	0.322	0.32
	*p*	<0.0001		<0.0001	<0.0001	<0.0001	0.7431	<0.0001	<0.0001	<0.0001	0.1871	<0.0001	0.0201	<0.0001	<0.0001	<0.0001	<0.0001
MPR	*r*	−0.355	−0.238		−0.106	−0.061	−0.121	−0.175	−0.033	−0.011	−0.116	0.039	−0.103	−0.093	−0.087	−0.006	−0.13
	*p*	<0.0001	<0.0001		0.0241	0.1942	0.0097	0.0002	0.4904	0.8139	0.0139	0.4096	0.0286	0.0491	0.0658	0.89	<0.01
FG	*r*	0.326	0.304	−0.106		0.157	0.013	−0.004	−0.051	−0.107	0.227	−0.04	0.169	0.159	0.096	0.104	0.029
	*p*	<0.0001	<0.0001	0.0241		<0.0001	0.7234	0.9238	0.1822	0.0051	<0.0001	0.3001	<0.0001	<0.0001	0.0116	0.007	0.45
BMI (kg/m2)	*r*	0.105	0.377	−0.061	0.157		−0.176	−0.029	−0.445	−0.556	0.181	−0.449	0.294	0.492	0.402	0.587	0.523
	*p*	0.0058	<0.0001	0.1942	<0.0001		<0.0001	0.4451	<0.0001	<0.0001	<0.0001	<0.0001	<0.0001	<0.0001	<0.0001	<0.0001	<0.0001
LH (mIU/mL)	*r*	0.188	0.012	−0.121	0.013	−0.176		0.069	0.091	0.106	−0.004	0.172	−0.084	−0.133	−0.065	−0.156	−0.174
	*p*	<0.0001	0.7431	0.0097	0.7234	<0.0001		0.0715	0.0169	0.0056	0.9082	<0.0001	0.027	0.0005	0.088	<0.0001	<0.0001
DHEA-S (umol/L)	*r*	0.518	0.366	−0.175	−0.004		0.069		0.013	0.023	−0.062	−0.002	−0.068	−0.028	−0.051	−0.0023	−0.018
	*p*	<0.0001	<0.0001	0.0002	0.9238	0.4451	0.0715		0.7247	0.5467	0.1057	0.961	0.0764	0.4696	0.1815	0.54	0.63
FGIR	*r*	0.001	−0.204	−0.033	−0.051	−0.445	0.091	0.013		0.903	−0.087	0.326	−0.176	−0.319	−0.281	−0.94	−0.48
	*p*	0.97	<0.0001	0.4904	0.1822	<0.0001	0.0169	0.7247		<0.0001	0.0232	<0.0001	<0.0001	<0.0001	<0.0001	<0.0001	<0.0001
QUICKI	*r*	−0.021	−0.271	−0.011	−0.107	−0.556	0.106	0.023	0.903		−0.146	0.386	−0.239	−0.408	−0.379	−1	−0.507
	*p*	0.5914	<0.0001	0.8139	0.0051	<0.0001	0.0056	0.5467	<0.0001		0.0001	<0.0001	<0.0001	<0.0001	<0.0001	<0.0001	<0.0001
Chol (mg/dL)	*r*	0.054	0.05	−0.116	0.227	0.181	−0.004	−0.062	−0.087	−0.146		0.124	0.849	0.463	0.342	0.166	0.196
	*p*	0.1577	0.1871	0.0139	<0.0001	<0.0001	0.9082	0.1057	0.0232	0.0001		0.0011	<0.0001	<0.0001	<0.0001	<0.0001	<0.0001
HDL (mg/dL)	*r*	−0.017	−0.266	0.039	−0.04	−0.449	0.172	−0.002	0.326	0.386	0.124		−0.232	−0.752	−0.437	−0.428	−0.777
	*p*	0.6614	<0.0001	0.4096	0.3001	<0.0001	<0.0001	0.961	<0.0001	<0.0001	0.0011		<0.0001	<0.0001	<0.0001	0.0001	<0.001
LDL (mg/dL)	*r*	0.012	0.089	−0.103	0.169	0.294	−0.084	−0.068	−0.176	−0.239	0.849	−0.232		0.676	0.298	0.27	0.362
	*p*	0.7474	0.0201	0.0286	<0.0001	<0.0001	0.027	0.0764	<0.0001	<0.0001	<0.0001	<0.0001		<0.0001	<0.0001	<0.0001	<0.0001
TC/HDL	*r*	0.061	0.271	−0.093	0.159	0.492	−0.133	−0.028	−0.319	−0.408	0.463	−0.752	0.676		0.611	0.489	0.815
	*p*	0.1111	<0.0001	0.0491	<0.0001	<0.0001	0.0005	0.4696	<0.0001	<0.0001	<0.0001	<0.0001	<0.0001		<0.0001	<0.0001	<0.0001
TG (mg/dL)	*r*	0.046	0.258	−0.087	0.096	0.402	−0.065	−0.051	−0.281	−0.379	0.342	−0.437	0.298	0.611		0.445	0.916
	*p*	0.2247	<0.0001	0.0658	0.0116	<0.0001	0.088	0.1815	<0.0001	<0.0001	<0.0001	<0.0001	<0.0001	<0.0001		<0.0001	<0.0001
HOMA-IR	*r*	0.05	0.322	−0.006	0.104	0.587	−0.156	−0.0023	−0.94	−1	0.166	−0.428	0.27	0.489	0.445		0.504
	*p*	0.19	<0.0001	0.89	0.007	<0.0001	<0.0001	0.54	<0.0001	<0.0001	<0.0001	0.0001	<0.0001	<0.0001	<0.0001		<0.0001
TG/HDL	*r*	0.035	0.32	−0.13	0.029	0.523	−0.174	−0.018	−0.48	−0.507	0.196	−0.777	0.362	0.815	0.916	0.504	
	*p*	0.36	<0.0001	<0.01	0.45	<0.0001	<0.0001	0.63	<0.0001	<0.0001	<0.0001	<0.001	<0.0001	<0.0001	<0.0001	<0.0001	

# Spearman rank correlation coefficient. Abbreviations: BMI—body mass index; TC—cholesterol; FGIR—fasting glucose to insulin ratio; FTI—free testosterone index; HDL—high-density lipoprotein cholesterol; HOMA-IR—Homeostatic Model Assessment for Insulin Resistance; LDL—low-density lipoprotein cholesterol; QUICKI—quantitative insulin sensitivity check index; TC—total cholesterol; TC/HDL—total cholesterol to high-density lipoprotein cholesterol ratio; TG—triglycerides; T—total testosterone.

**Table 4 ijerph-17-09291-t004:** Correlation between studied parameters in Group 1.

Variable	#	T	FTI	BMI	FGIR	QUICKI	Chol	HDL	LDL	TC/HDL	TG	HOMA-IR	TG/HDL
T (ng/mL)	*r*		0.586	0.093	−0.088	−0.059	−0.006	−0.1	0.051	0.098	0.062	0.065	0.077
	*p*		<0.0001	0.1163	0.1315	0.3116	0.9232	0.0856	0.3791	0.0932	0.2901	0.2672	0.1857
FTI	*r*	0.586		0.433	−0.417	−0.417	0.076	−0.428	0.22	0.458	0.294	0.419	0.395
	*p*	<0.0001		<0.0001	<0.0001	<0.0001	0.199	<0.0001	0.0002	<0.0001	<0.0001	<0.0001	<0.0001
BMI (kg/m2)	*r*	0.093	0.433		−0.518	−0.545	0.235	−0.461	0.383	0.586	0.425	0.548	0.511
	*p*	0.1163	<0.0001		<0.0001	<0.0001	0.0001	<0.0001	<0.0001	<0.0001	<0.0001	<0.0001	<0.0001
FGIR	*r*	−0.088	−0.417	−0.518		0.945	−0.106	0.439	−0.247	−0.481	−0.368	−0.945	−0.451
	*p*	0.1315	<0.0001	<0.0001		<0.0001	0.0706	<0.0001	<0.0001	<0.0001	<0.0001	<0.0001	<0.0001
QUICKI	*r*	−0.059	−0.417	−0.545	0.945		−0.148	0.447	−0.29	−0.513	−0.408	−1	−0.48
	*p*	0.3116	<0.0001	<0.0001	<0.0001		0.0116	<0.0001	<0.0001	<0.0001	<0.0001	<0.0001	<0.0001
TC (mg/dL)	*r*	−0.006	0.076	0.235	−0.106	−0.148		0.22	0.839	0.398	0.358	0.163	0.152
	*p*	0.9232	0.199	0.0001	0.0706	0.0116		0.0001	<0.0001	<0.0001	<0.0001	0.0052	0.0092
HDL (mg/dL)	*r*	−0.1	−0.428	−0.461	0.439	0.447	0.22		−0.161	−0.778	−0.449	−0.44	−0.763
	*p*	0.0856	<0.0001	<0.0001	<0.0001	<0.0001	0.0001		0.0057	<0.0001	<0.0001	<0.0001	<0.0001
LDL (mg/dL)	*r*	0.051	0.22	0.383	−0.247	−0.29	0.839	−0.161		0.686	0.384	0.303	0.351
	*p*	0.3791	0.0002	<0.0001	<0.0001	<0.0001	<0.0001	0.0057		<0.0001	<0.0001	<0.0001	<0.0001
TC/HDL	*r*	0.098	0.458	0.586	−0.481	−0.513	0.398	−0.778	0.686		0.636	0.518	0.805
	*p*	0.0932	<0.0001	<0.0001	<0.0001	<0.0001	<0.0001	<0.0001	<0.0001		<0.0001	<0.0001	<0.0001
TG (mg/dL)	*r*	0.062	0.294	0.425	−0.368	−0.408	0.358	−0.449	0.384	0.636		0.403	0.905
	*p*	0.2901	<0.0001	<0.0001	<0.0001	<0.0001	<0.0001	<0.0001	<0.0001	<0.0001		<0.0001	<0.0001
HOMA-IR	*r*	0.065	0.419	0.548	−0.945	−1	0.163	−0.44	0.303	0.518	0.403		0.475
	*p*	0.2672	<0.0001	<0.0001	<0.0001	<0.0001	0.0052	<0.0001	<0.0001	<0.0001	<0.0001		<0.0001
TG/HDL	*r*	0.077	0.395	0.511	−0.451	−0.48	0.152	−0.763	0.351	0.805	0.905	0.475	
	*p*	0.1857	<0.0001	<0.0001	<0.0001	<0.0001	0.0092	<0.0001	<0.0001	<0.0001	<0.0001	<0.0001	

# Spearman rank correlation coefficient. Abbreviations: BMI—body mass index; TC—cholesterol; FGIR—fasting glucose to insulin ratio; FTI—free testosterone index; HDL—high-density lipoprotein cholesterol; HOMA-IR—Homeostatic Model Assessment for Insulin Resistance; LDL—low-density lipoprotein cholesterol; QUICKI—quantitative insulin sensitivity check index; TC—total cholesterol; TC/HDL—total cholesterol to high-density lipoprotein cholesterol ratio; TG—triglycerides; T—total testosterone.

**Table 5 ijerph-17-09291-t005:** Correlation between studied parameters in Group 2.

Variable	#	BMI	FGIR	QUICKI	HDL	LDL	TC/HDL	TG	HOMA-IR	TG/HDL
BMI (kg/m2)	*r*		−0.578	−0.621	−0.451	0.246	0.515	0.48	0.605	0.535
	*p*		<0.0001	<0.0001	<0.0001	0.0001	<0.0001	<0.0001	<0.0001	<0.0001
FGIR	*r*	−0.578		0.953	0.373	−0.253	−0.451	−0.466	−0.952	−0.495
	*p*	<0.0001		<0.0001	<0.0001	<0.0001	<0.0001	<0.0001	<0.0001	<0.0001
QUICKI	*r*	−0.621	0.953		0.385	−0.23	−0.451	−0.496	−1	−0.521
	*p*	<0.0001	<0.0001		<0.0001	0.0002	<0.0001	<0.0001	<0.0001	<0.0001
TC (mg/dL)	*r*	0.158	−0.175	−0.155	0.132	0.8	0.374	0.331	0.137	0.158
	*p*	0.0115	0.0052	0.0134	0.0348	<0.0001	<0.0001	<0.0001	0.0291	0.0112
HDL (mg/dL)	*r*	−0.451	0.373	0.385		−0.244	−0.841	−0.48	−0.368	−0.783
	*p*	<0.0001	<0.0001	<0.0001		0.0001	<0.0001	<0.0001	<0.0001	<0.0001
LDL (mg/dL)	*r*	0.246	−0.253	−0.23	−0.244		0.678	0.309	0.229	0.334
	*p*	0.0001	<0.0001	0.0002	0.0001		<0.0001	<0.0001	0.0002	<0.0001
TC/HDL	*r*	0.515	−0.451	−0.451	−0.841	0.678		0.616	0.445	0.813
	*p*	<0.0001	<0.0001	<0.0001	<0.0001	<0.0001		<0.0001	<0.0001	<0.0001
TG (mg/dL)	*r*	0.48	−0.466	−0.496	−0.48	0.309	0.616		0.484	0.909
	*p*	<0.0001	<0.0001	<0.0001	<0.0001	<0.0001	<0.0001		<0.0001	<0.0001
HOMA-IR	*r*	0.605	−0.952	−1	−0.368	0.229	0.445	0.484		0.519
	*p*	<0.0001	<0.0001	<0.0001	<0.0001	0.0002	<0.0001	<0.0001		<0.0001
TG/HDL	*r*	0.535	−0.495	−0.521	−0.783	0.334	0.813	0.909	0.519	
	*p*	<0.0001	<0.0001	<0.0001	<0.0001	<0.0001	<0.0001	<0.0001	<0.0001	

# Spearman rank correlation coefficient. Abbreviations: BMI—body mass index; TC—cholesterol; FGIR-fasting glucose to insulin ratio; FTI—free testosterone index; HDL—high-density lipoprotein cholesterol; HOMA-IR—Homeostatic Model Assessment for Insulin Resistance; LDL—low-density lipoprotein cholesterol; QUICKI—quantitative insulin sensitivity check index; TC—total cholesterol; TC/HDL—total cholesterol to high-density lipoprotein cholesterol ratio; TG—triglycerides; T—total testosterone.

**Table 6 ijerph-17-09291-t006:** Correlation between the studied parameters in Group 3.

Variable	#	BMI	FGIR	QUICKI	Chol	HDL	LDL	TC/HDL	TG	HOMA-IR	TG/HDL
BMI (kg/m2)	*r*		−0.647	−0.643	0.117	−0.59	0.283	0.544	0.431	0.637	0.544
	*p*		<0.0001	<0.0001	0.1733	<0.0001	0.0008	<0.0001	<0.0001	<0.0001	<0.0001
FGIR	t	−0.647		0.908	−0.145	0.486	−0.24	−0.453	−0.433	−0.907	−0.5
	*p*	<0.0001		<0.0001	0.093	<0.0001	0.0048	<0.0001	<0.0001	<0.0001	<0.0001
QUICKI	*r*	−0.643	0.908		−0.202	0.516	−0.278	−0.504	−0.459	−1	−0.528
	*p*	<0.0001	<0.0001		0.0183	<0.0001	0.0011	<0.0001	<0.0001	<0.0001	<0.0001
TC (mg/dL)	*r*	0.117	−0.145	−0.202		−0.062	0.883	0.567	0.449	0.202	0.334
	*p*	0.1733	0.093	0.0183		0.4683	<0.0001	<0.0001	<0.0001	0.0186	0.0001
HDL (mg/dL)	*r*	−0.59	0.486	0.516	−0.062		−0.347	−0.837	−0.547	−0.517	−0.798
	*p*	<0.0001	<0.0001	<0.0001	0.4683		<0.0001	<0.0001	<0.0001	<0.0001	<0.0001
LDL (mg/dL)	*r*	0.283	−0.24	−0.278	0.883	−0.347		0.749	0.43	0.277	0.443
	*p*	0.0008	0.0048	0.0011	<0.0001	<0.0001		<0.0001	<0.0001	0.0011	<0.0001
TC/HDL	*r*	0.544	−0.453	−0.504	0.567	−0.837	0.749		0.683	0.504	0.832
	*p*	<0.0001	<0.0001	<0.0001	<0.0001	<0.0001	<0.0001		<0.0001	<0.0001	<0.0001
TG (mg/dL)	*r*	0.431	−0.433	−0.459	0.449	−0.547	0.43	0.683		0.459	0.932
	*p*	<0.0001	<0.0001	<0.0001	<0.0001	<0.0001	<0.0001	<0.0001		<0.0001	<0.0001
HOMA-IR	*r*	0.637	−0.907	−1	0.202	−0.517	0.277	0.504	0.459		0.529
	*p*	<0.0001	<0.0001	<0.0001	0.0186	<0.0001	0.0011	<0.0001	<0.0001		<0.0001
TG/HDL	*r*	0.544	−0.5	−0.528	0.334	−0.798	0.443	0.832	0.932	0.529	
	*p*	<0.0001	<0.0001	<0.0001	0.0001	<0.0001	<0.0001	<0.0001	<0.0001	<0.0001	

# Spearman rank correlation coefficient. Abbreviations: BMI—body mass index; TC—cholesterol; FGIR—fasting glucose to insulin ratio; FTI—free testosterone index; HDL—high-density lipoprotein cholesterol; HOMA-IR—Homeostatic Model Assessment for Insulin Resistance; LDL—low-density lipoprotein cholesterol; QUICKI—quantitative insulin sensitivity check index; TC—total cholesterol; TC/HDL—total cholesterol to high-density lipoprotein cholesterol ratio; TG—triglycerides; T—total testosterone.

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
