# Peer review of "Heterogeneity of Endocrinologic and Metabolic Parameters in Reproductive Age Polycystic Ovary Syndrome (PCOS) Women Concerning the Severity of Hyperandrogenemia—A New Insight on Syndrome Pathogenesis"

_ijerph, 2020, doi:10.3390/ijerph17249291_

Round 1

Reviewer 1 Report

English grammar & spelling

Authors should revise English Grammar throughout all document:

  • Replace “we” for “was/were” on lines 70,79, 89, 94/95, and others. Please, check all manuscript.
  • Add missing comas
  • Avoid long sentences i.e. line 221/223.

Abstract

Rephrase conclusion. It is hard to understand/follow how it is at the moment.

Introduction

More references should be provided throughout the Introduction, specifically:

  • when talking about % of prevalence of PCOS. And also, please, indicate which criteria this prevalence is based on.
  • For the associated metabolic risks (T2DM, gestational diabetes, etc.). Reference provided only /covers cardiovascular diseases.

Line 87/88: not clear what authors are trying to say. Please rephrase.

Statistical analysis

Indicate how analysis was performed when data was not normally distributed.

Results

Line 104: not clear what “first analysis” refers to. Please clarify.

Table 1 legend: Indicate that values represent mean (SD).

Figure 1: quality of the image is really bad. Plot titles and axis can’t be read. Please change.

Line 142/143: Poor English sentence. Please rephrase/change.

Discussion

Authors should discuss about the limitation of not using euglycemic-hyperinsulinemic clamp as the gold-standard for measuring insulin resistance/sensitivity in this study.

Also, authors should consider reordering the discussion to make it easier to read and follow. I suggest to move paragraph starting on line 306 after line 258.

Line 206: References provided are not appropriate to support prevalence of IR. Please, change. I suggest citing these papers: Stepto et al., Human Reprod 2013; Moghetti et al., JCEM 2013.

Line 207: provide reference for Azziz

Line 209/210: Please replace “Moreover, hyperinsulinemia by inhibiting the hepatic synthesis of sex hormone-binding globulin (SHBG) increases bio-available serum testosterone” for “Moreover, hyperinsulinemia increases bio-available serum testosterone by inhibiting the hepatic synthesis of sex hormone-binding globulin (SHBG)”

Line 221/223: Sentence is too long and it is hard to read. Please, change.

Line 285/287: Sentence is not linked to previous paragraph. Please remove and introduce concept within the next paragraph.

Author Response

Below I enlose the response to the Reviewer number 1

1. English grammar & spelling

Authors should revise English Grammar throughout all document:

  • Replace “we” for “was/were” on lines 70,79, 89, 94/95, and others. Please, check all manuscripts.
  • Add missing comas
  • Avoid long sentences, i.e., line 221/223.

We did revise the article upon vital advice. We tried to correct all the spelling and missing commas and other interpunction.

  1. Abstract Rephrase conclusion. It is hard to understand/follow how it is at the moment.

Thank You for this comment; the preliminary version's conclusion was indeed hard to read and understand. We did rephrase it from:

Conclusions No correlation between hyperandrogenemia and insulin sensitivity, dyslipidemia in women with high/very high androgenemia, which may suggest the importance of abnormalities in steroidogenesis and hyperinsulinemia in PCOS pathogenesis in this subpopulation. Metabolic abnormalities may contribute more significantly to PCOS development.

Into:

Conclusions: We found no correlation between testosterone levels and insulin sensitivity or dyslipidemia in women with PCOS, which may indicate the impact of steroidogenesis abnormalities and hyperinsulinemia on PCOS pathogenesis. Metabolic abnormalities may contribute more significantly than hyperandrogenemia to PCOS development.

  1. Introduction

More references should be provided throughout the Introduction, expressly:

  • when talking about % of prevalence of PCOS. And also, please, indicate which criteria this prevalence is based on.

Thank You for the remark. We did widen the description of PCOS prevalence, with the specification of different criteria used by various societies.

  • For the associated metabolic risks (T2DM, gestational diabetes, etc.). Reference provided only /covers cardiovascular diseases.

Thank You for the remark. We added citation according to the described features.

Line 87/88: not clear what authors are trying to say. Please rephrase.

We did rephrase the sentence into: The level of testosterone in our laboratory, that is proposed to be within the normal range for women of reproductive age, is 0.06–0.82 ng/mL.

Statistical analysis

Indicate how analysis was performed when data was not normally distributed.

We did recalculate our data once again, and in fact, most of the variables did not have the normal distribution (Kologomorow –Smirnov and d’Agostino Pearson tests). We recalculated table 1 using the Kruskal-Wallis test and presented the data as median (IQR 25-75). Table 2 was calculated correctly using Spearman’s rank correlation coefficient, but tables 3-5 had to be once again recalculated. We used Spearman’s rank correlation coefficient; the corrected data and tables are enclosed in the new version of the manuscript.

  1. Results

Line 104: not clear what “first analysis” refers to. Please clarify.

This characteristic initially did not indicate the relation between the patients' androgenemia and metabolic status in these subgroups of PCOS patients.

Table 1 legend: Indicate that values represent mean (SD).

We changed the whole table and indicated that the values represent median (IQR 25-75)

Figure 1: quality of the image is really bad. Plot titles and axis can’t be read. Please change.

We added each figure separately so that the resolution is better.

Line 142/143: Poor English sentence. Please rephrase/change.

We did change the sentence, so we hope it will be easier to read and understand.

  1. Discussion

Authors should discuss about the limitation of not using euglycemic-hyperinsulinemic clamp as the gold-standard for measuring insulin resistance/sensitivity in this study.

We did add this as a limitation of the study at the end of the papers.

Also, authors should consider reordering the discussion to make it easier to read and follow. I suggest moving paragraph starting on line 306 after line 258.

We did change the order of the paragraphs. Thank You for that suggestion.

Concerning the second reviewer's opinion, we also shortened some parts of the discussion.

Line 206: References provided are not appropriate to support prevalence of IR. Please, change. I suggest citing these papers: Stepto et al., Human Reprod 2013; Moghetti et al., JCEM 2013.

Thank You for that comment and the indication of proper references. We added them to the paper.

Line 207: provide reference for Azziz

We added the missing reference.

Line 209/210: Please replace “Moreover, hyperinsulinemia by inhibiting the hepatic synthesis of sex hormone-binding globulin (SHBG) increases bio-available serum testosterone” for “Moreover, hyperinsulinemia increases bio-available serum testosterone by inhibiting the hepatic synthesis of sex hormone-binding globulin (SHBG)”

We did replace the sentence according to Your suggestion.

Line 221/223: Sentence is too long and it is hard to read. Please, change.

We divided the sentence into two separate sentences.

Line 285/287: Sentence is not linked to previous paragraph. Please remove and introduce concept within the next paragraph.

Reviewer 2 Report

Review of “Heterogeneity of Endocrinologic and Metabolic Parameters in
Reproductive Age Polycystic Ovary Syndrome (PCOS) Women Concerning the
Severity of Hyperandrogenemia—A New Insight on Syndrome Pathogenesis” (ijerph-1009351)

This study investigated the association among metabolic parameters in patients with PCOS.

This study is potentially interesting; however, several problems to be solved.

  1. In this study, all continues variables shown normal distributions? I think not all variables were normal distributions, the authors should use Mann–Whitney U test, Spearman's rank correlation coefficient or Kruskal-Wallis tests.
  2. Please add all participants data (Table 1).
  3. Table 1. Please add the FBS, insulin. Furthermore, how about the HOMA-IR?
  4. In addition, TG/HDL is reported to be associated with insulin resistance (Fukuda Y, et al. Liver Int. 2016;36(5):713-720.). How about the association between TG/HDL and testosterone?
  5. How about the data of free testosterone ?
  6. Figure 1. The resolution of Figure 1 is low and difficult to understand.
  7. Results and Discussion. Table 3-Table 5. Why this sub-group analyses were performed and what add the new findings from this sub-group analyses?
  8. Results of Table 3-5 are too long and not reader friendly.
  9. In addition, Discussion is also too long.

Author Response

Thank You for Your revision. I here enlclose the answear, comments to Your review.

  1. In this study, all continues variables shown normal distributions? I think not all variables were normal distributions, the authors should use Mann–Whitney U test, Spearman's rank correlation coefficient or Kruskal-Wallis tests.

This is a significant comment from the reviewer. We did recalculate our data once again. In fact, most of the variables did not have a normal distribution (Kologomorow –Smirnov and d’Agostino Pearson tests). We recalculated table 1 using the Kruskal-Wallis test and presented them as median (IQR 25-75). Table 2 was calculated correctly using Spearman’s rank correlation coefficient, but table 3-5 were once again recalculated also using Spearman’s rank correlation coefficient

  1. Please add all participants data (Table 1).

We do not fully understand this part of the revision. We added the full number of participants ( n... ) in all three groups. If the reviewer wishes, we may provide the whole excel table with data concerning each separate data.

  1. Table 1. Please add the FBS, insulin. Furthermore, how about the HOMA-IR?

Thank You for the comment. We overlooked those data in the previous version. We added FBS, insulin, and HOMA-IR results into the table.

  1. In addition, TG/HDL is reported to be associated with insulin resistance (Fukuda Y, et al. Liver Int. 2016;36(5):713-720.). How about the association between TG/HDL and testosterone?

We added this ratio and calculated the correlation. However, we did not see any significant correlation between this parameter and testosterone. Once again, it seems that the levels of testosterone do not correlate with metabolic parameters.

  1. How about the data of free testosterone ?

We did not measure levels of free testosterone.

  1. Figure 1. The resolution of Figure 1 is low and difficult to understand.

We added each figure separately, so it has a better resolution.

  1. Results and Discussion. Table 3-Table 5. Why these sub-group analyses were performed and what add the new findings from this sub-group analyses?

The sub group analyses were performed, because we thought, that it may help to prove that metabolic parameters and their changes in PCOS patients are not associated with hormonal differences, mainly testosterone levels. That higher or lower level of testosterone does not influence the severity or changes of metabolic complications.

  1. Results of Table 3-5 are too long and not reader friendly.

We did narrow the results and tables 3-5. In the revised version, we presented only those results that chad correlation bigger than (0.5) or (-0.5) so it is shorter and more transparent for the reader. If the reviewer suggests that those data are not crucial for the paper, maybe we can transfer those tables and its description into supplementary material.

  1. In addition, discussion is also too long.

Thank You for the comment. We did shorten the discussion.

Round 2

Reviewer 1 Report

Introduction. Provide prevalence of IR in women with PCOS, as requested in previous comments.

Line 199. I suggest removing "or even is the main reason for" and just leave " hyperinsulinemia is a major contributor to the overproduction of male sex steroids" as this is a still major point of debate.

Line 285. I believe authors are speculating too much on this sentence "This method's usage could indicate an even bigger occurrence of insulin resistance, therefore indicate stronger correlations between insulin and studies parameters" and therefore I suggest to rephrase it.

Conclusions section is too long, please reduce.

Author Response

  1. Provide prevalence of IR in women with PCOS, as requested in previous comments.

 We added the the prevalence of IR in the introduction section.

  1. Line 199.I suggest removing "or even is the main reason for" and just leave " hyperinsulinemia is a major contributor to the overproduction of male sex steroids" as this is a still major point of debate.

Thank You for this comment, we did change it according to Your suggesiton.

  1. Line 285.I believe authors are speculating too much on this sentence "This method's usage could indicate an even bigger occurrence of insulin resistance, therefore indicate stronger correlations between insulin and studies parameters" and therefore I suggest to rephrase it.

We removed this sentence, leaving just the annotation about the limitation of the study, because of the lack of not using the hyperinsulinemic-euglycemic clamp in the study.

Not using the hyperinsulinemic-euglycemic clamp is the study's limitation when speculating about hyperinsulinemia effect on the variables studied in this paper. However, the study of Pisprasert et al. indicated that insulin sensitivity indices based on glucose and insulin levels work well in the studies where we examine a homogenous group of patients (same gender, same race population)

  1. Conclusions section is too long, please reduce.

We moved the sentences aobut advanteges and disadvantages to the end of discussion. We additionally shortened the conclusion part to the most relevant informations.

Reviewer 2 Report

The authors revised well. Only question is below.

Please add a table of unify all participants not divided into groups. 

Author Response

Please add a table of unify all participants not divided into groups. 

We enclosed the missing table, as the reviewer requested.